# Evaluation of a very brief pedometer-based physical activity intervention delivered in NHS Health Checks in England: The VBI randomised controlled trial

Wendy Hardeman[1]*, Joanna Mitchell[2], Sally Pears[2], Miranda Van Emmenis[2], Florence Theil[2], Vijay S. Gc[3,4], Joana C. Vasconcelos[5], Kate Westgate[6], Søren Brage[6], Marc Suhrcke[4,7], Simon J. Griffin[6,8], Ann Louise Kinmonth[8], Edward C. F. Wilson[3,9], A. Toby Prevost[5], Stephen Sutton[2], on behalf of the VBI Research Team¶

1 Behavioural and Implementation Science Group, School of Health Sciences, University of East Anglia, Norwich Research Park, Norwich, United Kingdom, 2 Behavioural Science Group, Primary Care Unit, Department of Public Health and Primary Care, Cambridge Institute of Public Health, Cambridge, United Kingdom, 3 Health Economics Group, Norwich Medical School, University of East Anglia, Norwich, United Kingdom, 4 Centre for Health Economics, University of York, York, United Kingdom, 5 Imperial Clinical Trials Unit, Imperial College London, London, United Kingdom, 6 MRC Epidemiology Unit, University of Cambridge, Cambridge, United Kingdom, 7 Luxembourg Institute of Socio-Economic Research, Esch-sur-Alzette, Luxembourg, 8 Primary Care Unit, Department of Public Health and Primary Care, Cambridge Institute of Public Health, Cambridge, United Kingdom, 9 Cambridge Centre for Health Services Research, Department of Public Health and Primary Care, Cambridge Institute of Public Health, Cambridge, United Kingdom

¶ Membership of the VBI Research Team is provided in the Acknowledgments.
* W.Hardeman@uea.ac.uk

**Data Availability Statement:** Anonymised patient-level data will be made available on request to the University of Cambridge Primary Care Unit

## Abstract

### Background

The majority of people do not achieve recommended levels of physical activity. There is a need for effective, scalable interventions to promote activity. Self-monitoring by pedometer is a potentially suitable strategy. We assessed the effectiveness and cost-effectiveness of a very brief (5-minute) pedometer-based intervention ('Step It Up') delivered as part of National Health Service (NHS) Health Checks in primary care.

### Methods and findings

The Very Brief Intervention (VBI) Trial was a two parallel-group, randomised controlled trial (RCT) with 3-month follow-up, conducted in 23 primary care practices in the East of England.

Participants were 1,007 healthy adults aged 40 to 74 years eligible for an NHS Health Check. They were randomly allocated (1:1) using a web-based tool between October 1, 2014, and December 31, 2015, to either intervention (505) or control group (502), stratified by primary care practice. Participants were aware of study group allocation. Control participants received the NHS Health Check only. Intervention participants additionally received Step It Up: a 5-minute face-to-face discussion, written materials, pedometer, and step chart.

Secretariat (PCUPA@medschl.cam.ac.uk) after submitting a proposal and signing a data access agreement. Consent for data sharing was not explicitly obtained from participants, but the data are anonymised, and the risk of identification is low.

**Funding:** This article presents independent research funded by the National Institute for Health Research (NIHR) under the Programme Grants for Applied Research programme (Grant Reference Number RP-PG-0608-10079). The views expressed in this publication are those of the author(s) and not necessarily those of the NIHR or the Department of Health and Social Care. The funder had no role in study design, data collection, data analysis, data interpretation, the writing of the manuscript, and decision to submit the manuscript for publication. The researchers and funders were independent. The VBI team acknowledges the support of the National Institute for Health Research Clinical Research Network (NIHR CRN). The work of KW and SB was funded by the UK Medical Research Council (MC_UU_12015/3) and the NIHR Biomedical Research Centre Cambridge [IS-BRC-1215-20014]. SJG is NIHR Senior Investigator. The University of Cambridge has received salary support in respect of SJG from the NHS in the East of England through the Clinical Academic Reserve.

**Competing interests:** All authors have read the journal's policy and have declared that no competing interests exist.

**Abbreviations:** BCT, behaviour change technique; cpm, counts per minute; CRF, case report form; CVD, cardiovascular disease; GCSE, General Certificate of Secondary Education; IMD, Index of Multiple Deprivation; ITT, Intention to Treat; MAR, Missing at Random; NHS, National Health Service; PACE-Lift, Pedometer Accelerometer Consultation Evaluation-Lift; PACE-UP, Pedometer And Consultation Evaluation trial; PAEE, physical activity energy expenditure; PP, Per Protocol; RCT, randomised controlled trial; RPAQ, Recent Physical Activity Questionnaire; VBI, very brief intervention; WPAI, Work Productivity and Activity Impairment.

The primary outcome was accelerometer-based physical activity volume at 3-month follow-up adjusted for sex, 5-year age group, and general practice. Secondary outcomes included time spent in different intensities of physical activity, self-reported physical activity, and economic measures. We conducted an in-depth fidelity assessment on a subsample of Health Check consultations.

Participants' mean age was 56 years, two-thirds were female, they were predominantly white, and two-thirds were in paid employment. The primary outcome was available in 859 (85.3%) participants. There was no significant between-group difference in activity volume at 3 months (adjusted intervention effect 8.8 counts per minute [cpm]; 95% CI −18.7 to 36.3; $p = 0.53$). We found no significant between-group differences in the secondary outcomes of step counts per day, time spent in moderate or vigorous activity, time spent in vigorous activity, and time spent in moderate-intensity activity (accelerometer-derived variables); as well as in total physical activity, home-based activity, work-based activity, leisure-based activity, commuting physical activity, and screen or TV time (self-reported physical activity variables). Of the 505 intervention participants, 491 (97%) received the Step it Up intervention. Analysis of 37 intervention consultations showed that 60% of Step it Up components were delivered faithfully. The intervention cost £18.04 per participant. Incremental cost to the NHS per 1,000-step increase per day was £96 and to society was £239. Adverse events were reported by 5 intervention participants (of which 2 were serious) and 5 control participants (of which 2 were serious). The study's limitations include a participation rate of 16% and low return of audiotapes by practices for fidelity assessment.

## Conclusions

In this large well-conducted trial, we found no evidence of effect of a plausible very brief pedometer intervention embedded in NHS Health Checks on objectively measured activity at 3-month follow-up.

## Trial registration

Current Controlled Trials (ISRCTN72691150).

## Author summary

### Why was this study done?

- Systematic reviews support the effectiveness and cost-effectiveness of brief (up to 30 minutes) physical activity interventions in primary care and the effectiveness of intensive pedometer interventions.

- There is a need for briefer interventions in primary care, where time is limited but potential reach is large.

- However, the reviews show uncertainty about the effectiveness and cost-effectiveness of very brief (up to 5 minutes) interventions in primary care.

- Until now, there have been no randomised controlled trials (RCTs) of a very brief pedometer-based intervention to increase physical activity in primary care.

## What did the researchers do and find?

- We randomly assigned participants who attended preventive health checks in primary care (National Health Service [NHS] Health Checks) to the health check alone or additionally a very brief pedometer-based physical activity intervention delivered by practice nurses or healthcare assistants. We measured physical activity objectively at 3 months after the intervention.

- Our large trial found no benefit of a very brief physical activity intervention in the context of preventive health checks in primary care.

- Despite the intervention being apparently simple and very brief, fidelity of delivery was suboptimal. Trial participants were more active than might have been expected.

- The economic evaluation shows a small added cost for a small and uncertain benefit.

## What do these findings mean?

- The absence of a positive effect of a very brief physical activity intervention challenges the commissioning of such interventions in this context.

- Primary care practitioners should continue to opportunistically provide very brief advice about physical activity.

## Introduction

The global prevalence of self-reported inactivity among adults—defined as less than 150 minutes of moderate-intensity physical activity per week, or less than 75 minutes of vigorous-intensity aerobic activity per week, or an equivalent combination of the two—is 23% [1]. Less than half of United Kingdom adults report meeting recommended physical activity levels. When measured by accelerometer, only 5% achieve recommended levels [2]. Worldwide, physical inactivity is responsible for 6% of coronary heart disease, 7% of type 2 diabetes, 10% of breast cancer, 10% of colon cancer, 9% of all-cause mortality [3], 3.8% of dementia cases [1], and 13.4 million disability-adjusted life years [4]. The global cost of physical inactivity was estimated at US$67.5 billion (£46.9 billion, €51.0 billion) in 2013 [4], and the direct cost to the UK National Health Service (NHS) was estimated to be £0.9 billion in 2006–2007 (US$1.34 billion, €1.1 billion, converted at 2007 Purchasing Power Parity) [5].

In healthcare settings, very brief interventions (VBIs), defined as deliverable within 5 minutes [6], are promising: they are potentially scalable and can reach many people cheaply with limited time requirement for health practitioners. Very brief opportunistic behaviour change advice in health and care settings is advocated, for instance, by the Department of Health and Social Care for England and local government [7,8]. Evidence supports the effectiveness [9] and cost-effectiveness [10] of brief interventions of up to 30 minutes (albeit based on self-reported physical activity), but there is uncertainty about VBIs. Meta-analyses have shown that pedometer interventions can increase walking by 2,000 to 2,500 steps per day [11,12]. In a preliminary trial. we evaluated 3 VBIs: a pedometer-based VBI showed most promise in terms of potential efficacy, feasibility, acceptability, and cost [13].

We undertook to (1) estimate the effectiveness of a very brief pedometer-based intervention ('Step it Up') in increasing objectively measured physical activity in adults aged 40 to 74 years attending NHS Health Checks in primary care (a cardiovascular disease [CVD] risk reduction programme) compared with the Health Check alone, (2) estimate its cost-effectiveness compared with the Health Check alone from the perspectives of the NHS and society, and (3) assess the fidelity of delivery and mechanisms underlying any intervention effects.

## Methods

### Design

This was a two parallel-group, randomised controlled trial (RCT) with 1:1 individual allocation. The study was registered before participant recruitment, and the protocol has been published [14]. The trial compared the Step it Up intervention delivered in an NHS Health Check (intervention group) to the NHS Health Check alone (usual care control group). Follow-up was at 3 months after the Health Check. The study was approved by the East of England-Cambridge East Research Ethics Committee (14/EE/1004). The trial is registered with Current Controlled Trials, number ISRCTN72691150.

### Participants

We recruited patients who were eligible for the NHS Health Check, aged between 40 and 74 years, with no diagnosis of vascular disease and not on a care pathway for known risk factors (e.g., raised blood pressure) [15]. We excluded patients unable to provide written informed consent (e.g., insufficient grasp of the English language) and patients whose General Practitioner considered them unsuitable for inclusion.

Participants were recruited from 23 general practices in urban and rural areas across the East of England: 12 practices in Cambridgeshire, 8 in Hertfordshire and Bedfordshire, and 3 in Norfolk. All participants gave written informed consent.

### Randomisation and masking

Each practice was given instructions to randomly select a subsample of 250 eligible patients out of all eligible patients on the database, and two practices randomly selected further subsamples. Individual randomisation was stratified by primary care practice. The allocation ratio was 1:1 with randomly permuted blocks of sizes 2, 4, and 6 to ensure even randomisation and low predictability of assignment within each stratum. Practice staff (nurse or healthcare assistant) randomised the participants at the start of the Health Check using a web-based tool (www.sealedenvelope.com). Practitioners and participants were blind to allocation until this point. The statistical analysis plan was completed prior to the receipt of outcome data for analysis and is available as S1 Text.

### Procedures

Healthcare practitioners attended a 3-hour training session on study procedures and intervention delivery at their practice. To promote fidelity of delivery, practitioners were encouraged to use a brief procedure, a case report form (CRF), and the Step it Up intervention booklet during the consultation.

Participant recruitment was through the NHS Health Check programme [16]. Participants were invited to the NHS Health Check by mail and were encouraged to make an appointment with their practice and to mention their interest in taking part in the trial.

At the start of the Health Check, the health practitioner obtained informed consent from participants following Good Clinical Practice guidelines [17]. Patients who consented were asked to complete a short questionnaire used to characterise the sample and were randomised. The remainder of the consultation was the usual NHS Health Check (control group) and, additionally, Step it Up for intervention participants. Practitioners were asked to use the CRF as a prompt and a written record of the consultation, which included assessment of self-reported baseline physical activity level [18].

Participants in the control group received the usual NHS Health Check only. This included blood pressure measurement, calculation of BMI from measured height and weight, and collection of a blood sample [19]. This information was used to calculate the patient's modelled CVD risk over the next 10 years and guide the offer of appropriate support. Intervention participants received the NHS Health Check, followed by Step it Up, which had been carefully developed and tested for feasibility, fidelity, acceptability, and potential efficacy [13]. Step it Up consisted of a 5-minute face-to-face discussion, provision of a Yamax Digiwalker SW200 pedometer (Polygon Direct Ltd, UK), a Step Chart for self-monitoring, and a Step it Up Booklet. The key behaviour change techniques (BCTs) [20] included in Step it Up were goal setting, action planning, feedback, and self-monitoring of behaviour [14]. Participants were told that 10,000 steps per day was a good target to aim for but were encouraged to gradually increase their step goals each week if they were able to. They were given a Step Chart and encouraged to use this to set weekly step goals and record daily steps. The health practitioners did not suggest specific activities but mentioned that any activity that raises heart rate, makes you breathe a little more heavily, and causes a light sweat counts as physical activity.

## Measures

Baseline data were collected by practitioners. Outcomes were measured by accelerometer and questionnaires 3 months later, which were returned by participants in reply-paid envelopes [14].

The primary outcome was physical activity (total volume of body movement) measured by tri-axial accelerometry (Actigraph GT3X+ or Actigraph w-GT3X-BT, Actigraph, Pensacola, Florida) expressed as average vector magnitude acceleration (counts per minute [cpm]). Participants were posted an accelerometer on an elastic belt and asked to wear it around their waist on the right hip for 7 consecutive days during waking hours. Data were collected at 60 Hz and integrated into 10-second epochs; we removed non-wear time defined as $\geq$90 minutes of consecutive zeros (on the vertical axis) and summarised remaining vector magnitude data into average acceleration (cpm). Data were considered valid if there were at least 3 days of data, each day requiring $\geq$600 minutes of wear time. Secondary physical activity outcomes derived from accelerometer data were step counts (average step counts per day, derived from frequency analysis) as well as average number of minutes per day spent in moderate activity (2,690–6,166 cpm), vigorous activity ($\geq$6,167 cpm), and moderate or vigorous activity ($\geq$2,690 cpm) [21]. We did not analyse sedentary/light intensity activity (<2,690 cpm) because it represented an internal category of the variable that would not have a practical interpretation. We collected self-reported physical activity data using the validated Recent Physical Activity Questionnaire (RPAQ) [22], which participants completed following accelerometer wear.

We assessed primary and secondary NHS care contacts and out-of-pocket expenditure on health, sports clubs, or other physical activities via a bespoke questionnaire. Work place productivity was based on an adapted version of the validated Work Productivity and Activity Impairment (WPAI) Questionnaire [23].

Process measures in the same questionnaire included recall of the Health Check consultation and enactment of key BCTs included in Step it Up such as goal setting and self-

monitoring. To assess fidelity, we selected a random sample of 5 NHS Health Check consultations (10%, intervention and control group) from each practice for audio recording. We obtained 63 audio recordings from 13 out of 23 practices: 37 intervention and 26 control consultations. All consultations were independently doubly coded. Interrater reliability was >75% for all but one item, so we used the data of the main coder for analysis. We decided a priori that levels above 60% constituted an acceptable level of fidelity and decided during data analysis on a contamination level of 30% for control participants receiving intervention components.

## Statistical analysis

We aimed to recruit a sample size of 1,140 (570 per group) to follow up 394 participants per group after 3 months, allowing for missing or incomplete accelerometer data for 30% of randomised participants. This provided 80% power to detect a 0.2-SD ('small') difference in mean activity between groups (40 cpm), based on the SD of 200 cpm estimated in our preliminary trial [13] (alpha = 0.05, two-sided test). As attrition was lower than anticipated at 15%, we could recruit fewer participants in accordance with our protocol [14].

We conducted our analyses according to Consolidated Standards of Reporting Trials (CONSORT) guidelines (S1 Table) and agreed the analysis plan a priori with the independent Trial Steering Committee [14]. We used analysis of covariance to test for intervention effects on continuous outcomes and quantified these with differences in means and 95% CIs, adjusting for primary care practice, sex, and age. We used an Intention to Treat (ITT) approach, supported by a Per Protocol (PP) analysis for the primary outcome analysis. The PP population comprised the subset of the ITT population who received their randomised intervention as determined from the CRF completed by the practitioners. All significance tests were two-sided and assessed at the 5% level of significance, and secondary outcome results were interpreted cautiously due to multiple comparisons.

Missing data for the primary outcome were handled within a sensitivity analysis. This examined the robustness of the main analysis result to its 'Missing at Random' (MAR) assumption by exploring the impact of departures from this assumption on the primary outcome results [24]. We examined a set of optimistic and pessimistic scenarios for the intervention effect size in participants with missing data by prespecifying a range for accelerometer cpm from −50 cpm to +50 cpm over which the mean of the 'unobserved outcome data' in a group might depart (or be different) from the mean of the 'observed outcome data' in that group. Three scenarios were examined in the sensitivity analysis. These reflect whether departures from the MAR assumption applied within the intervention group only, within the control group only, or within both groups equally and in the same direction (thereby potentially cancelling out across the sensitivity range, if the dropout rate were to be the same in both groups).

We examined prespecified subgroup variables in relation to the primary outcome: baseline CVD risk, sex, age (40–59, 60–74 years), ethnic group, educational qualifications, employment status, household income, marital status, home ownership, vehicle ownership, a deprivation score (calculated by scoring one point for each of the following: no qualifications, unemployed or full-time student, renting their home, no cars), and the Index of Multiple Deprivation (IMD) 2007, derived from the participant's home postal code. For continuous potential moderators, e.g., modelled 10-year cardiovascular risk, we compared the estimated intervention effect in the highest tertile to that in the combined lower and middle tertiles of the moderator. SPSS version 23.0 (IBM Corporation, Armonk, NY) was used for data checking and analyses.

We conducted a within-trial economic evaluation from the perspectives of the NHS (comprising costs of delivering Step it Up and patient NHS contacts) and society (defined as the

sum of NHS cost, personal out of pocket expenditure, and morbidity-related lost productivity), reporting incremental cost per 1,000 extra steps per day. Full details are reported in S2 Text.

We used descriptive analyses for mechanisms of any intervention effect and fidelity.

## Patient and public involvement

The trial had strong patient and public involvement from a panel of 4 members, two of whom were also Trial Steering Committee members. They commented on and contributed to the trial protocol, informed consent procedures and use of the web-based randomisation tool, the patient information sheet and consent form, procedures to increase recruitment and retention rates, and the contents and delivery of the Step it Up intervention, follow-up questionnaire, and the findings and dissemination. We have disseminated the results to our participants.

## Results

Based on data from 19 out of 23 practices, 6,200 participants were invited and 1,057 assessed for eligibility (see Fig 1). Four practices did not keep records on numbers of participants invited and assessed. We recruited 1,007 eligible participants (control: 502, intervention: 505) from 23 general practices in the East of England between 1 October 2014 and 31 December 2015. Eleven control participants and 14 intervention participants did not receive the allocated intervention due to practitioner error. Of randomised participants, 85.3% provided valid data on the primary outcome at 3-month follow-up (control: 88% [442]; intervention: 83% [417]; $p = 0.014$).

Participant baseline characteristics were similar in each group (Table 1). Mean age was 56 years, and two-thirds of participants were female. They were predominantly white. Two-thirds were in paid employment, and 85% reported having a General Certificate of Secondary Education (GCSE), A-level, or degree-level qualification. Thirty-one percent of participants reported being inactive or moderately inactive.

Accelerometer cpm at 3-month follow-up (Table 2 and Fig 2) were similar in the control and intervention groups (660 versus 668 cpm, respectively). This corresponds to an adjusted intervention effect of 8.8 cpm (95% CI −18.7 to 36.3).

### Sensitivity analyses

**Sensitivity of the result to missing data.** Compared to those with missing primary outcome data, participants who completed follow-up with primary outcome had higher modelled 10-year CVD risk (median 6.7% [IQR 2.7%–11.5%] versus 4.9% [1.9%–8.7%], $p = 0.005$), were less likely to be in paid employment (59% versus 75%, $p < 0.001$) and manual workers (25% versus 41%, $p = 0.002$), and were older (56.6 years [SD = 9.4] versus 52.9 years [SD = 9.4]). When adjusting the primary outcome model for other covariates as a post hoc sensitivity analysis, the results remained nonsignificant (the intervention effect ranged from 5.2 to 11.5) (see S2 Table).

**Sensitivity of the results to MAR assumption.** In the first sensitivity analysis scenario assuming that participants with unobserved outcome data would take mean values from −50 cpm to +50 cpm from the adjusted observed effect, the resulting intervention effect ranged from 2.8 to 14.8, reflecting departures from the MAR assumption within the intervention group only. For departures within the control group only (scenario 2), the resulting intervention effect ranged from 0.1 to 17.5. For departures within both groups equally and in the same direction (scenario 3), the resulting effect ranged from 6.1 to 11.5. All values within these ranges represent a positive nonsignificant intervention effect and indicate that the main result is robust.

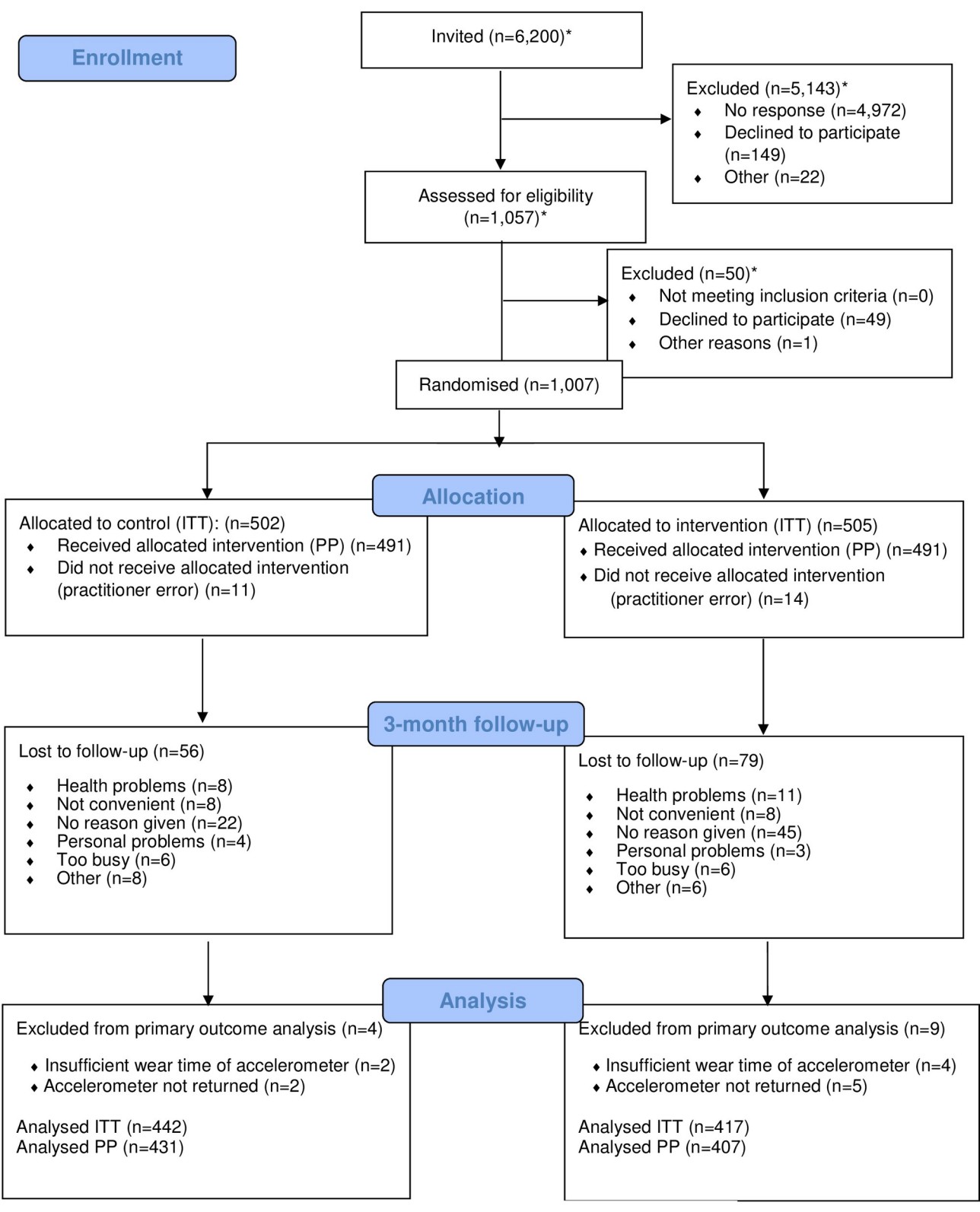

**Fig 1. Flow of participants through the trial comparing the NHS Health Check plus a very brief pedometer-based intervention with the NHS Health Check only.** ITT, Intention to Treat; NHS, National Health Service; PP, Per Protocol.

**Table 1. Baseline characteristics of participants allocated to intervention and control groups.** Values are percentages (numbers) unless otherwise stated.

| Characteristics | Control (*n* = 502) | Intervention (*n* = 505) |
|---|---|---|
| **Mean (SD) age** | 56.5 (9.4) | 55.7 (9.6) |
| **Female** | 61% (305) | 63% (316) |
| **White ethnicity** | 95% (476/500) | 96% (484/504) |
| **Married or cohabiting** | 81% (375/465) | 80% (383/480) |
| **Have dependants** | 35% (164/468) | 39% (186/482) |
| **Work status** | *n* = 472 | *n* = 482 |
| Paid work | 61% (286) | 62% (301) |
| Unemployed/homemaker | 6% (29) | 6% (28) |
| Full-time student | 0% (0) | 0% (1) |
| Retired | 32% (153) | 31% (148) |
| Other | 1% (4) | 1% (4) |
| **Income** | *n* = 410 | *n* = 424 |
| Less than £18,000 | 26% (105) | 21% (88) |
| £18,000–£30,999 | 22% (91) | 22% (94) |
| £31,000–£51,999 | 28% (114) | 29% (124) |
| £52,000–£100,000 | 18% (72) | 20% (85) |
| Greater than £100,000 | 7% (28) | 8% (33) |
| **Occupational group** | *n* = 295 | *n* = 314 |
| Manual | 24% (71) | 27% (84) |
| Nonmanual | 68% (200) | 65% (203) |
| Other | 8% (24) | 9% (27) |
| **Highest qualification** | *n* = 485 | *n* = 494 |
| None | 9% (46) | 9% (44) |
| GCSE | 60% (290) | 66% (326) |
| A-level | 6% (30) | 5% (26) |
| Degree | 19% (91) | 15% (76) |
| Other | 6% (28) | 4% (22) |
| **Accommodation** | *n* = 464 | *n* = 482 |
| Ownership | 86% (399) | 88% (423) |
| Rent | 13% (59) | 11% (51) |
| Other | 1% (6) | 2% (8) |
| **Car ownership** | 94% (442/469) | 95% (458/483) |
| **Deprivation score**[*] | *n* = 450 | *n* = 473 |
| 0 | 76% (341) | 81% (384) |
| 1 | 19% (84) | 14% (65) |
| 2 | 5% (23) | 4% (18) |
| 3 | 0.2% (1) | 1% (3) |
| 4 | 0.2% (1) | 1% (3) |
| **Median (IQR) IMD**[**] | *n* = 502 | *n* = 505 |
|  | 11.1 (5.7–18.9) | 10.9 (5.9–18.0) |
| **Median (IQR) CVD risk score** | *n* = 487 | *n* = 498 |
|  | 6.45% (2.76%–11.34%) | 6.30% (2.40%–10.89%) |
| **Physical activity status (GPPAQ)** | *n* = 502 | *n* = 505 |
| Inactive | 12.5% (63) | 13.7% (69) |
| Moderately inactive | 19.3% (97) | 16.0% (81) |
| Moderately active | 35.1% (176) | 35.2% (178) |

(*Continued*)

**Table 1.** (Continued)

| Characteristics | Control (*n* = 502) | Intervention (*n* = 505) |
|---|---|---|
| Active | 33.1% (166) | 35.0% (177) |

*Calculated by scoring one point for each of the following: no qualifications, unemployed or full-time student, renting their home, no cars.

**Derived from the postal code recorded on the consent form.

**Abbreviations:** CVD, cardiovascular disease; GCSE, General Certificate of Secondary Education; GPPAQ, General Practice Physical Activity Questionnaire; IMD, Index of Multiple Deprivation; IQR, interquartile range

**PP analysis.** The PP analysis showed no difference in conclusion from the ITT results: data differed in only 2.7% of the participants. The adjusted intervention effect was 9.1 cpm (95% CI −18.6 to 36.7; *p* = 0.52).

We found no evidence for heterogeneity of effect between subgroups, except for modelled CVD risk score. The intervention was associated with a reduction in activity among those at highest CVD risk and an increase in activity among those at lowest CVD risk (*p* = 0.002) (Table 3).

For secondary outcomes, there were no significant differences between groups in accelerometer-derived step counts per day and time spent in moderate or vigorous, vigorous, and moderate-intensity activity (Table 2 and Fig 2). Similarly, we found no significant differences between groups in self-reported total physical activity, home-based activity, work-based activity, leisure-based activity, commuting physical activity, and screen or TV time (Table 4 and Fig 3).

The intervention cost £18.04 per participant (£11.25 pedometer, £4.67 face-to-face consultation time with nurse, £2.12 materials). Other NHS costs were not statistically significantly different between groups (mean [SE] = +£21.55 [£24.21]). Total societal costs (including intervention, NHS, personal out-of-pocket and lost-productivity costs) were not significantly different either (mean [SE] = +£53.46 [£76.97]). Point estimate incremental cost per 1,000 steps per day was £96.32 from the NHS and £238.89 from the societal perspectives.

Based on practitioner records, 491 of 505 participants received the Step it Up intervention as planned (Fig 1). Table 5 shows the process evaluation findings. At 3-month follow-up, significantly more intervention than control participants recalled receiving physical activity advice and intervention materials and reported enacting BCTs (e.g., using a pedometer to count their steps) in daily life.

The in-depth fidelity assessment revealed that contamination was minimal: intervention components were delivered in more than 30% of control group sessions for 3 components only, related to discussing physical activity recommendations [13]. This was included in routine Health Checks in several practices, constituting variation in routine care rather than contamination.

An estimate of fidelity of intervention delivery based on an in-depth assessment of 37 of 505 (7.3%) audio-recorded consultations was 60%, meaning that—on average—9 out of 15 components were delivered. Components that were relatively poorly delivered concerned the BCTs: giving feedback on physical activity (51.4%), mentioning the effectiveness of pedometers (18.9%), and prompting goal setting (21.6%). Components to promote participant engagement were also relatively poorly delivered, e.g., asking participants whether they were aware of the physical activity recommendations (29.7%) and whether they had any questions (18.9%). There was large variability in fidelity of delivery across practices and practitioners. The average

**Table 2. Objectively measured physical activity by trial group and between-group differences at 3-month follow-up.**

| Variable | Control | | Intervention | | Intervention relative to control | |
|---|---|---|---|---|---|---|
| | N | Mean (95% CI)* | N | Mean (95% CI)* | Comparison of means** | p-Value |
| Total physical activity volume (cpm)*** | 442 | 660 (641 to 679) | 417 | 668 (648 to 689) | 8.8 (−18.7 to 36.3) | 0.53 |
| Step counts per day | 442 | 8,191 (7,911 to 8,471) | 417 | 8,419 (8,110 to 8,729) | 242 (−172 to 656) | 0.25 |
| Time (min/d) in moderate or vigorous activity | 442 | 76.7 (73.5 to 80.1) | 417 | 77.3 (73.9 to 80.9) | 0.9% (−4.9% to 7.2%) | 0.76 |
| Time (min/d) in vigorous activity | 442 | 2.9 (2.6 to 3.2) | 417 | 3.2 (2.9 to 3.6) | 11.9% (−2.9% to 28.8%) | 0.12 |
| Time (min/d) in moderate activity | 442 | 71.8 (68.9 to 74.8) | 417 | 72.0 (68.8 to 75.2) | 0.3% (−5.4% to 6.5%) | 0.91 |

*Means are geometric means for time in activity at different intensities and compared as the percentage by which the intervention group's geometric mean is raised relative to that in the control group after adjustment for covariates.

**Comparison of means is adjusted for sex, 5-year age group, and practice.

***These are vector magnitude cpm.

**Abbreviation:** cpm, counts per minute

duration of the Health Check was a mean (SD) of 15 minutes and 16 seconds (6 minutes and 24 seconds). The average duration of VBI delivery was a mean (SD) of 3 minutes and 9 seconds (1 minute and 51 seconds).

During the trial, 10 participants reported adverse events, 5 in the intervention and 5 in the control group. Four of these were serious adverse events, 1 in the intervention group (stitches to a head injury after falling while cycling) and 2 in the control group (sepsis and surgery after hospital admission, and being fitted with a pacemaker after experiencing blackouts, respectively). The fourth serious adverse event was a suspected unexpected serious adverse event: an intervention participant reported having a pacemaker fitted between the health check and follow-up assessment but had not been diagnosed with a pre-existing condition.

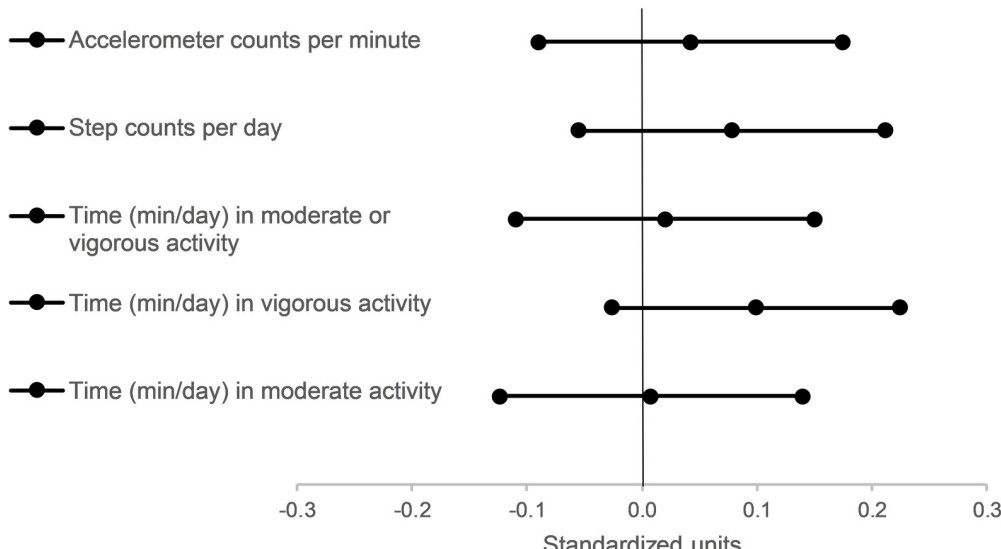

**Fig 2. Differences in objectively measured physical activity between intervention and control groups at 3-month follow-up (expressed in standardised units and 95% CIs).**

**Table 3. Subgroup analyses of primary outcome.**

| Variable | Subgroup 1 | | Subgroup 2 | | Test between subgroups |
|---|---|---|---|---|---|
| | Intervention–Control Mean (95% CI) | | Intervention–Control Mean (95% CI) | | |
| | *N* | Difference (95% CI) | *N* | Difference (95% CI) | *p*-Value |
| **Sex** | 529 | Females | 330 | Males | 0.35 |
| | | 19.3 (−15.9 to 54.5) | | −7.9 (−52.4 to 36.6) | |
| **Age** | 369 | 60 to 74 years | 490 | 40 to 59 years | 0.24 |
| | | −10.3 (−52.6 to 32.0) | | 23.2 (−13.4 to 59.8) | |
| **Deprivation score** | 163 | Score > 0 | 624 | Score = 0 | 0.50 |
| | | 34.6 (−30.4 to 99.5) | | 9.7 (−23.0 to 42.3) | |
| **Education** | 224 | Other | 610 | None or GCSE | 0.95 |
| | | 11.3 (−44.2 to 66.8) | | 9.1 (−23.9 to 42.0) | |
| **Marital status** | 650 | Married/cohabiting | 159 | Single | 0.53 |
| | | 4.6 (−27.4 to 36.7) | | 28.0 (−37.5 to 93.4) | |
| **Paid work** | 332 | No paid work | 482 | Paid work | 0.54 |
| | | −3.7 (−48.0 to 40.6) | | 14.5 (−22.2 to 51.3) | |
| **Occupation**[†] | 348 | Not manual/other | 117 | Manual | 0.23 |
| | | 10.3 (−31.6 to 52.1) | | 62.1 (−11.3 to 135.5) | |
| **Income** | 322 | <£31,000 | 387 | ≥£31,000 | 0.35 |
| | | −6.5 (−52.5 to 39.6) | | 23.7 (−18.1 to 65.5) | |
| **CVD risk** | 546 | Lower and middle tertiles | 294 | Upper tertile | |
| | | 44.0 (9.3 to 78.7) | | −48.7 (−96.1 to −1.2) | 0.002 |
| **IMD** | 578 | Lower and middle tertiles | 281 | Upper tertile | |
| | | −3.0 (−36.6 to 30.5) | | 36.0 (−12.4 to 84.4) | 0.19 |
| **GPPAQ**[†] | 261 | Inactive/moderately inactive | 598 | Moderately active/active | |
| | | 13.6 (−36.0 to 63.2) | | 2.1 (−30.4 to 34.7) | 0.71 |

[†]This was not prespecified.

**Abbreviations:** CVD, cardiovascular disease; GCSE, General Certificate of Secondary Education; GPPAQ, General Practice Physical Activity Questionnaire; IMD, Index of Multiple Deprivation

**Table 4. Self-reported physical activity by study group and between-group differences at 3-month follow-up.**

| Variable | Control | | Intervention | | Intervention relative to control | |
|---|---|---|---|---|---|---|
| | *N* | Mean (95% CI)* | *N* | Mean (95% CI)* | Comparison of means** | *p*-Value |
| **PAEE (kJ/kg/d)** | 440 | 28.0 (26.0 to 30.0) | 418 | 29.5 (27.5 to 31.7) | 5.4% (−4.2% to 16.0%) | 0.28 |
| **Home-based PAEE (kJ/kg/d)** | 439 | 2.7 (2.5 to 2.9) | 418 | 2.9 (2.7 to 3.1) | 6.3% (−5.3% to 19.3%) | 0.30 |
| **Work-based PAEE (kJ/kg/d)** | 273 | 11.8 (10.6 to 13.2) | 269 | 13.3 (11.8 to 15.0) | 9.0% (−6.5% to 27.1%) | 0.27 |
| **Leisure-based PAEE (kJ/kg/d)** | 440 | 12.0 (10.7 to 13.4) | 416 | 12.0 (10.8 to 13.4) | 0.7% (−13.7% to 17.5%) | 0.93 |
| **Commuting PAEE (kJ/kg/d)** | 266 | 0.6 (0.5 to 0.8) | 257 | 0.6 (0.4 to 0.7) | −10.0% (−34.0% to 22.6%) | 0.50 |
| **Screen/TV time (h/d)** | 439 | 2.8 (2.6 to 2.9) | 418 | 2.8 (2.6 to 2.9) | 0.005 (−0.2 to 0.2) | 0.96 |

*Means are geometric means for skewed PAEE outcomes and compared as the percentage by which the intervention group's geometric mean is raised relative to that in the control group after adjustment for covariates.

**Comparison of means is adjusted for sex, 5-year age group, and practice.

**Abbreviation:** PAEE, physical activity energy expenditure

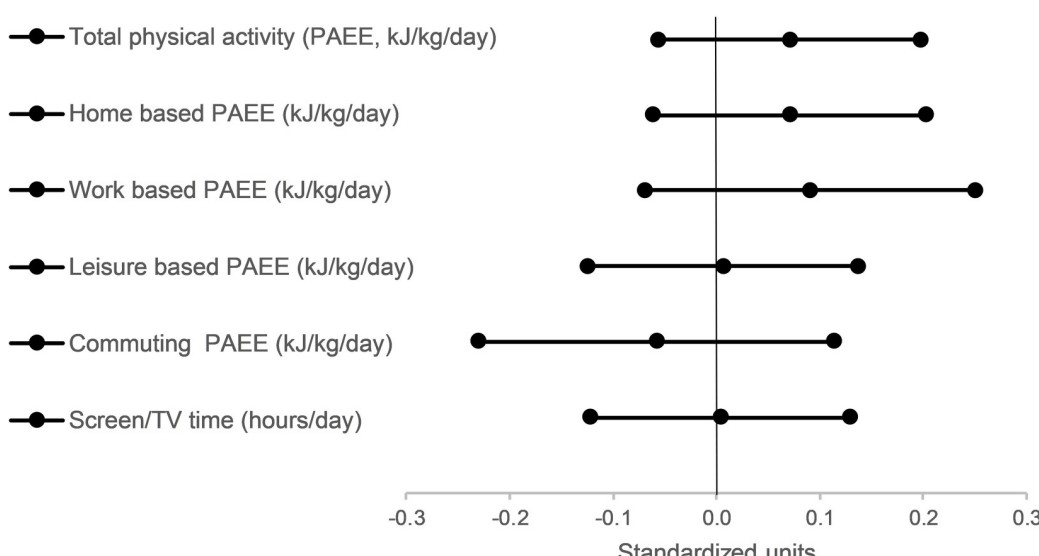

**Fig 3. Differences in self-reported physical activity measures between intervention and control groups at 3-month follow-up (expressed in standardised units and 95% CIs).** PAEE, physical activity energy expenditure.

**Table 5. Recall of the NHS Health Check, enactment of BCTs, and contamination.** Values are percentages (numbers).

| | Control | Intervention | *p*-Value |
|---|---|---|---|
| | *n* = 442[*] | *n* = 420[*] | |
| Physical activity was mentioned in my Health Check | 82% (359/437) | 94% (392/415) | *p* < 0.001 |
| I was given advice about how to become more active | 39% (168/434) | 72% (296/410) | *p* < 0.001 |
| At my Health Check, I was given a booklet called Step it Up and a pedometer | 6% (27/437) | 93% (379/406) | *p* < 0.001 |
| If yes, I read the booklet[**] | 51% (28/55) | 90% (346/385) | *p* < 0.001 |
| If yes, I still have the booklet | 37% (20/54) | 80% (306/383) | *p* < 0.001 |
| Since my Health Check, I have used a pedometer to count steps | 15% (64/435) | 88% (368/417) | *p* < 0.001 |
| Since my Health Check, I have written down my step counts | 5% (24/437) | 68% (281/415) | *p* < 0.001 |
| Since my Health Check, I have set myself goals to increase my physical activity | 31% (134/435) | 49% (202/415) | *p* < 0.001 |
| Since my Health Check, I have thought about the benefits of physical activity | 82% (357/437) | 90% (376/416) | *p* < 0.001 |
| I know how much physical activity I should do per week according to government recommendations | 43% (190/438) | 64% (266/417) | *p* < 0.001 |
| Since my Health Check, I have become more aware of how physically active I am | 66% (289/436) | 82% (341/415) | *p* < 0.001 |
| I know of someone else who has taken part in the study | 6% (24/433) | 9% (39/418) | *p* = 0.035 |
| If yes, they were in the Intervention group (A) or the Control group (B) | A: 47% (8/17) | A: 44% (14/32) | *p* = 0.83 |
| | B: 53% (9/17) | B: 56% (18/32) | |

[*]12% (*n* = 60) of the control group and 17% (*n* = 85) of the intervention group had missing questionnaire data.

[**]Some participants reported having read and/or kept the booklet or pedometer when they reported not receiving it. This is not a data entry error; potential reasons are participant error or interpreting the question as meaning something else. For instance, participants may have received other written materials during the NHS Health Check.

**Abbreviations:** BCT, behaviour change technique; NHS, National Health Service

## Discussion

Our trial showed no evidence for a positive effect of a 5-minute pedometer-based intervention delivered as part of NHS Health Checks in primary care on objectively measured and self-reported physical activity after 3 months, compared to the Health Check alone. Sensitivity analysis indicated that the main result is robust. Step it Up resulted in an increase of about 9 accelerometer cpm, whereas we considered an increase of 40 counts—equivalent to a small effect size—to be a minimum clinically significant effect. We found no evidence for a positive intervention effect in any prespecified subgroups, except that the intervention appeared to be more effective among participants with lower modelled CVD risk and less so in those with higher risk. The cost per patient of delivering Step it Up was £18, a small amount and known with reasonable certainty. As anticipated, we observed no statistically significant difference in other costs.

### Interpretation

We discuss potential explanations for our findings in terms of study design, intervention, target group, and context. We demonstrated robust trial execution, so results are unlikely to be due to randomisation errors, confounding, selection bias, differential dropout, or contamination. The two groups were well matched at baseline and 3-month follow-up. Retention at follow-up was high at 85%, and any differences between those lost to follow-up and those who completed follow-up were similar enough in the two groups not to influence the main result. Any differences between groups at follow-up may have been attenuated by higher reactivity of control group participants to wearing an accelerometer as they had not previously received a pedometer. Such reactivity has been observed in previous physical activity trials [25].

Contamination was minimal and does not explain the findings. Suboptimal intervention delivery by practitioners, lack of participant engagement, and insufficient use of key recommended strategies in daily life are potential explanations. Setting physical activity goals is a key strategy, but at 3 months, only 49% of intervention participants—compared with 31% of control participants—reported setting physical activity goals. This may be due to the finding from the in-depth fidelity assessment that practitioners rarely prompted goal setting. Intervention duration was just over 3 minutes, 2 minutes less than planned (as estimated from the audio recordings) and shorter than the 5 minutes in our previous trial, in which fidelity of delivery of the pedometer intervention was higher at 72% [13]. Brief physical activity advice to control participants may have diluted any effect, but the fidelity assessment suggests that this did not happen. Rather, relatively poor delivery of the BCTs and strategies to increase participant engagement may explain the findings. The training in study procedures proved time-consuming and challenging for some practitioners, which may have affected their ability to master intervention delivery. Perhaps practitioners were rushed for time, lacked skills in engaging participants, or lacked confidence in delivering very brief advice [26]. Finally, the intervention content, which focused on self-monitoring using pedometers, may have been ineffective. However, meta-analyses have shown that more intensive pedometer interventions increase physical activity [11,12], and we selected and optimised the intervention after extensive development and pilot work [6,13].

Our target group was already relatively active, and there may have been limited capacity for a VBI to have an effect over and above the Health Check alone. At follow-up, objectively measured step counts in the control group (which provides the best estimate of the absolute physical activity distribution at baseline), though not accounting for any NHS Health Check effect, was 8,191 steps per day. This is slightly higher than observed in similar trials of pedometer-based interventions in UK primary care (7,479 in Pedometer And Consultation Evaluation

[PACE-UP] [27]) as well as higher than 7,380 in the control group and 7,314 in the intervention group of Pedometer Accelerometer Consultation Evaluation-Lift (PACE-Lift) [28]. Our intervention had a positive effect among participants with lower CVD risk and a negative effect in those with higher risk, although we interpret this cautiously given the inconsistent directions of effect, the number of subgroup tests, and the very small effect size. Other trials have not found significant associations between cardiovascular risk factors and change in physical activity [29,30].

Step it Up was delivered in a consultation focusing on vascular disease risk, which rendered physical activity benefits for disease prevention particularly salient, whereas our participants were on average 56 years and not on disease registers. Health Checks may not have been the optimal context, although participants in our previous studies mentioned that a discussion about physical activity during Health Checks was a good reminder [6].

The effectiveness of Step it Up may be increased by better delivery of BCTs and patient engagement in the consultation; offering repeated VBIs; referral to follow-up support delivered by digital technologies, by phone, or face to face; and by having an environment that is supportive of walking and cycling. This requires new research into their cost-effectiveness. Given current financial constraints in the NHS, practitioners could signpost apparently healthy adults of preretirement age to relatively cheap external support such as mobile phone applications or encourage them to purchase a pedometer. VBIs to promote physical activity might best be offered opportunistically to people with long-term conditions for which physical activity is an important component of disease management, as well as to older adults to halt the age-related decline in physical activity, alongside prescribing medication to reduce risk. A population-based study with repeated physical activity measures showed that middle-aged and older adults, including people with CVD and cancer, can live longer by becoming more physically active, irrespective of past activity levels and risk factors [31]. Physical inactivity is a habitual, environmentally cued behaviour. Inactivity is not simply a challenge for the health service but is a societal problem requiring a multisectoral, multidisciplinary public health response [32].

### Study strengths and limitations

A key strength of our trial was its internal validity. We had a sufficiently large sample size to be able to detect a small effect. Our sample was well-balanced, and retention at 3-month follow-up was high in both arms. We used a validated objective measure of physical activity and population-based sampling from primary care registers. The intervention was novel in terms of its brevity and inclusion of evidence-based BCTs and was based on extensive development and pilot work [6].

While it could be argued that absence of baseline measurement is a potential limitation, we believe this not to be the case. With a large sample size and high retention, comparison of values of a precise, objective physical activity measure between intervention and control groups gives a valid estimate of effect. Our estimate is also a fairly good measure of change, assuming that physical activity was similar in the two groups at baseline due to randomisation and follow-up too short for population shifts in the outcome. The main advantage of baseline measures is the improvement in precision due to correlation and ability to analyse group changes over time, but our large sample size provided sufficient precision. On the other hand, baseline measurement in a pragmatic trial of a VBI has several disadvantages. Measurement effects may preclude an intervention effect on behaviour [33–35]. Our pilot work showed that baseline measurement, which included objectively measured physical activity, reduced NHS Health Check uptake, which was not acceptable to participating practices as Health Checks constitute routine care [13]. Physical activity screening during the Health Check to exclude very active

adults prior to randomisation was not feasible as it would have added time and changed routine practice. Our participants were more active than might have been expected, potentially reducing room for improvement, although self-reported physical activity at baseline did not moderate any intervention effects.

External validity or generalisability from clinical trials is limited by participant profile. Sixteen percent (1,007/6,200) of adults invited agreed to take part and were randomised. This participation rate was similar to the PACE-UP trial, which used population-based sampling in a similar age group (10%) [27], and was lower than PACE-Lift (30%), which recruited among adults in the retirement age from practice records [28]. Our participants were representative of adults attending NHS Health Checks in the East of England. During 2015 and 2019, 47.4% of eligible people in England took up an NHS Health Check invitation [36]. The majority of adults aged 40 to 74 years are employed and may lack time for trial participation. Attrition was low overall, slightly greater in the intervention arm, but this was not found to influence the main finding. Finally, only 13 of 23 practices returned audiotapes, and observed fidelity levels may overestimate actual levels.

## Comparison with other studies

The UK PACE-Lift trial evaluated a 10-week pedometer-based intervention in primary care consisting of four 30-minute, individually tailored sessions with practice nurses and found a significant effect at 3 and 12 months (1,037 and 609 additional steps per day, respectively). The UK PACE-UP trial evaluated a 12-week pedometer-based intervention delivered by mail only and by mail plus three 20-minute practice nurse consultations [27]. Both interventions significantly increased step counts per day at 12 months (642 for the postal and 677 for the nurse intervention) and 3-year follow-up (627 for the postal and 670 for the nurse intervention) compared to the control group, which received usual care, with no significant additional benefit of the nurse intervention at either follow-up.

Our findings do not imply that primary care practitioners should refrain from providing (very) brief physical activity advice opportunistically as our evidence is limited to NHS Health Checks. They should be considered in the context of meta-analytic evidence supporting intensive pedometer interventions in predominantly small samples of younger participants [11,12], the potential scope for more faithful delivery of VBIs, 30-minute brief physical activity advice in primary care, more intensive pedometer-based interventions in primary care among slightly less active participants who received individualised walking plans, and contacts with practice nurses or research assistants.

## Conclusions

This large, well-conducted trial of a plausible very brief pedometer-based intervention embedded in NHS Health Checks found no evidence of effect on objectively measured activity at 3-month follow-up. Despite the intervention being apparently simple and very brief, fidelity of delivery was suboptimal. Trial participants were more active than might have been expected. Commissioners should consider effect size, context, population, risk, and opportunity cost in commissioning individual health-service–based preventive interventions. Continual adding of preventive actions of limited benefit to clinical encounters may overburden consultations and adversely impact the efficiency and effectiveness of primary care.

## Supporting information

**S1 Text. VBI trial statistical analysis plan.**
(PDF)

**S2 Text. Evaluation of a very brief pedometer-based physical activity intervention delivered in NHS Health Checks in England: The VBI randomised controlled trial.**
(DOCX)

**S1 Table. CONSORT.** Consolidated Standards of Reporting Trials (CONSORT) checklist.
(DOCX)

**S2 Table. Sensitivity analysis.**
(DOCX)

## Acknowledgments

This study was conducted on behalf of the Very Brief Interventions Programme team (see www.phpc.cam.ac.uk/pcu/research/research-projects-list/vbi/vbi-research-team for team members). We thank James Brimicombe for his guidance in data management, for designing the VBI trial database, and for his support throughout the data collection process. We thank Lewis Griffiths (MRC Epidemiology Unit) for support throughout the physical activity data collection. We thank the patients, nurses, healthcare assistants, general practitioners, practice managers, and administrative staff from the following general practices: Alconbury and Brampton Surgery; Aspland's Surgery, Woburn Sands; Bayfield Surgery, Docking; Bridge Street Medical Centre, Cambridge; Bridge Street Surgery, Downham Market; Church Street Surgery, Ware; Comberton and Eversden Surgery; De Parys Medical Centre, Bedford; Hanscombe House Surgery, Hertford; Lensfield Road Medical Practice, Cambridge; North Street Medical Practice, Peterborough; Parsonage Surgery, Bishop's Stortford; Pemberley Surgery, Bedford; Salisbury House Surgery, Leighton Buzzard; St John's Surgery, Terrington St John; The Acorn Surgery, Huntingdon; The Cornerstone Practice, March; The Old Exchange @ East Street, St. Ives; The Over Surgery, Over; The Riverside Practice, March; The Spinney Surgery, St. Ives; Wallace House Surgery, Hertford; and York St Medical Practice, Cambridge. We thank all members of the Trial Steering Committee, including Falko Sniehotta, Beelin Baxter, Jennifer Bostock, Carolyn Read, and Becky Reynolds, and all members of the Patient and Public Involvement (PPI) panel, including Becky Reynolds, Jennifer Bostock, Ian Gardner, Janet Gibson, and Francesco Palma.

## Author Contributions

**Conceptualization:** Wendy Hardeman, Sally Pears, Marc Suhrcke, Simon J. Griffin, Ann Louise Kinmonth, Edward C. F. Wilson, A. Toby Prevost, Stephen Sutton.

**Data curation:** Joanna Mitchell, Joana C. Vasconcelos, Kate Westgate.

**Formal analysis:** Vijay S. Gc, Joana C. Vasconcelos, Kate Westgate, Edward C. F. Wilson, A. Toby Prevost.

**Funding acquisition:** Wendy Hardeman, Marc Suhrcke, Simon J. Griffin, Ann Louise Kinmonth, Edward C. F. Wilson, A. Toby Prevost, Stephen Sutton.

**Investigation:** Wendy Hardeman, Joanna Mitchell, Sally Pears, Miranda Van Emmenis, Florence Theil, Vijay S. Gc, Edward C. F. Wilson, A. Toby Prevost, Stephen Sutton.

**Methodology:** Wendy Hardeman, Sally Pears, Vijay S. Gc, Joana C. Vasconcelos, Marc Suhrcke, Edward C. F. Wilson, A. Toby Prevost, Stephen Sutton.

**Project administration:** Wendy Hardeman, Joanna Mitchell, Edward C. F. Wilson, A. Toby Prevost, Stephen Sutton.

**Software:** Joanna Mitchell, Vijay S. Gc, Joana C. Vasconcelos, Edward C. F. Wilson, A. Toby Prevost.

**Supervision:** Wendy Hardeman, Marc Suhrcke, Ann Louise Kinmonth, Edward C. F. Wilson, A. Toby Prevost, Stephen Sutton.

**Validation:** Wendy Hardeman, Joanna Mitchell, Sally Pears, Vijay S. Gc, Joana C. Vasconcelos, Kate Westgate, Søren Brage, Edward C. F. Wilson, A. Toby Prevost, Stephen Sutton.

**Visualization:** Joana C. Vasconcelos.

**Writing – original draft:** Wendy Hardeman, Vijay S. Gc, Joana C. Vasconcelos, Edward C. F. Wilson, A. Toby Prevost, Stephen Sutton.

**Writing – review & editing:** Wendy Hardeman, Joanna Mitchell, Sally Pears, Miranda Van Emmenis, Florence Theil, Vijay S. Gc, Joana C. Vasconcelos, Kate Westgate, Søren Brage, Marc Suhrcke, Simon J. Griffin, Ann Louise Kinmonth, Edward C. F. Wilson, A. Toby Prevost, Stephen Sutton.

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
