## [Decision Letter · Decision Letter 0]

10 Oct 2019

Dear Dr. Hardeman,

Thank you very much for submitting your manuscript "Effectiveness and cost-effectiveness of a very brief pedometer-based physical activity intervention delivered in NHS Health Checks: The VBI randomised controlled trial" (PMEDICINE-D-19-02857) for consideration at PLOS Medicine. 

Your paper was discussed among the editorial team and sent to independent reviewers, including a statistical reviewer. The reviews are appended at the bottom of this email and any accompanying reviewer attachments can be seen via the link below:

[LINK]

In light of these reviews, we will not be able to accept the manuscript for publication in the journal in its current form, but we would like to invite you to submit a revised version that addresses the reviewers' and editors' comments fully. You will appreciate that we cannot make a decision about publication until we have seen the revised manuscript and your response, and we expect to seek re-review by one or more of the reviewers. 

We hope to receive your revised manuscript by Oct 31 2019 11:59PM. Please email us (plosmedicine@plos.org) if you have any questions or concerns.

Please let me know if you have any questions. Otherwise, we will look forward to receiving your revised manuscript in due course. 

Sincerely,

Richard Turner, PhD

rturner@plos.org

Our data policy calls for a primary non-author contact for inquiries about access to data, and so we ask you to remove the author contact details. Please adapt the data statement wording to "... will be made available upon reasonable request". It may be helpful to add additional institutional contact details at a future stage. 

It may be that the competing interest with AbbVie can be removed in view of the date: please consult our conflict of interest policy (https://journals.plos.org/plosmedicine/s/competing-interests). 

Please adapt the title to "Evaluation of a very brief ...". 

Please add an additional sentence or two to your abstract to summarize the secondary outcome findings. 

We ask you to add a new final sentence of the "methods and findings" subsection of your abstract to summarize the study's main limitations. 

Please refer to the attached CONSORT document at a suitable point in your methods section. We ask you to include the study analysis plan as a supplementary document if possible, and again refer to this in the main text. Please highlight any analyses that were not prespecified in the main text (including that noted in the supplementary file). 

At the start of the discussion section of your main text, "rigorously conducted" seems a little strident, and could be de-emphasized by being moved to the "strengths and limitations" section. 

There may be an indication in figure 1 and elsewhere of greater attrition in the intervention arm, and we suggest addressing this in the "strengths and limitations" section. 

Please substitute "sex" for "gender" as appropriate throughout the text. 

Throughout the text, please adapt reference call-outs to the following style: "... per day [11,12].".

Please review the reference list to ensure that journal names are abbreviated consistently (e.g., "PLoS Med." for reference 27). 

Please adapt your attached CONSORT checklist so that individual items are referred to by section (e.g., "Methods") and paragraph number rather than by line or page numbers, as the latter generally change upon publication. 

As the trial protocol is published, you may wish to remove the attached document. 

Comments from the reviewers:

*** Reviewer #1 (statistical reviewer): 

This is a well-conducted RCT on the effectiveness and cost-effectiveness of a very brief pedometer-based physical activity intervention (VBI) delivered in NHS Health Checks. The study design, sample size calculation, randomisation, statistical methods and analyses, and presentation (tables and figures) and interpretation of results are mostly adequate and of a good standard. However, there are still a few issues needing attention.

1) On page 17, it would be good to provide a table on comparisons between patients with complete data and those with missing data, maybe as supplementary tables.

2) Table 1. Could authors please put a percentage sign (%) for all the percentages in the table?

3) The first paragraph on page 20 provides key results for the paper including ITT analysis, sensitivity analysis and PP analysis. Could authors please highlight these analyses (maybe use subtitles) and also make it very clear whether these results are significant or not, therefore deliver a clear message on whether these analyses changed results in any way. Also, for sensitivity and PP analyses could authors please provide complete results as supplementary tables?

4) In table 2, the primary outcome analysis was only adjusted for age, gender and practice however there are quite a few other variables in the baseline which could potentially be adjusted such as Physical activity status and multiple deprivation (IMD). Would adjusting for more baseline variables change the final results in any way?

*** Reviewer #2: 

This is a well-written report of an apparently well-conducted trial addressing an important question. The interpretation of the results is appropriate. 

I support the choice of a pragmatic trial in a "real-world" context. The findings are of course disappointing but this manuscript nonetheless makes an important contribution.

My only concern is the statement "Fidelity of delivery was partial." in the abstract. I think this is potentially misleading given the information on page 26 that "491/505 participants received the 'Step It Up' intervention as planned." I understand that this statement is based on the assessment of program content but feel that this should be made clearer and both pieces of information should be presented in the abstract. 

*** Reviewer #3: 

This is an interesting topic and novel, useful and pragmatic approach, with potential for wide public health reach. It demonstrates a good example of the integration of BCTs within very short intervention.

There are a number of issues which I think need to be explained more clearly in the paper before publishing:

* Most participants were already active at baseline (GPPAQ and report of control group's step count >8000 steps), and so it is unclear why they received an intervention designed for inactive people and were not screened out. A sub-group analysis on the inactive patients may be more appropriate to demonstrate the effectiveness of the intervention. 

* Author Summary statement - last point "Our general population findings do not imply that primary care practitioner should refrain from providing (very) brief PA advice opportunistically" has too many double negatives. Simply - Primary Care professionals should continue to providing very brief PA advice, opportunistically.

* Given the focus on a pragmatic intervention and approach, one of the key findings should be in relation to the poor fidelity of the intervention - it is important to note that the intervention was not delivered as intended on this occasion. This is especially pertinent given the aspects which were not delivered relate to aspects expected to support behaviour change. The results of the fidelity test should be in the results section of the abstract prior to the sentence "There were no significant between-group differences in activity volume (8.8 counts per minute; 95% CI -18.7, 36.3)" in order to alert readers that they should interpret the lack of difference between groups with care. 

* Re: NHS Health Checks - would be useful to know how many people attend these across England, to better understand reach. 

* Although there is a protocol paper available - Information is insufficient to understand intervention - An additional sentence here would be useful - What goals were suggested? Were they individualised? Over how many weeks? Etc

***

[LINK]

---

## [Editor Report · Decision Letter 1]

16 Jan 2020

Dear Dr. Hardeman,

Thank you very much for re-submitting your manuscript "Evaluation of a very brief pedometer-based physical activity intervention delivered in NHS Health Checks: The VBI randomised controlled trial" (PMEDICINE-D-19-02857R1) for consideration at PLOS Medicine.

I have discussed the paper with editorial colleagues and our academic editor, and I am pleased to tell you that, provided the remaining editorial and production issues are dealt with, we expect to be able to accept the paper for publication in the journal.

[LINK]

Please let me know if you have any questions. Otherwise, we look forward to receiving the revised manuscript shortly. 

Sincerely,

Richard Turner, PhD

rturner@plos.org

Requests from Editors:

Please remove the word "reasonable" from the data access statement (we feel that this is unnecessary in that the process for acquiring data is described). 

Please adapt the title to indicate the country, e.g., "... NHS Health Checks in England ...".

To the "methods and findings" subsection of your abstract, please add brief aggregate demographic characteristics of study participants. 

Please remove the sentence beginning "The statistical analysis plan was completed ..." from the abstract to save space. 

Please add the relevant p value alongside the CI to the primary endpoint difference quoted in the abstract. 

Regarding adverse events, please make that "... of which two were serious". 

At the start of the discussion section of the main text, please adapt the wording to "... an increase of about 9 accelerometer ...". 

In the same paragraph, regarding the subgroup analyses we ask you to soften the wording to "... the intervention appeared to be more effective among participants ...". 

Please ensure that all entries in the reference list meet journal format. All text in italics should be converted to plain text; and where appropriate 6 rather than 3 author names should be listed, followed by "et al.". Please abbreviate journal names consistently (e.g., "PLoS Med."). 

Please substitute a version of the attached CONSORT checklist in which the tracking has been cleared. 

***

---

## [Editor Report · Decision Letter 2]

31 Jan 2020

Dear Dr Hardeman, 

On behalf of my colleagues and the academic editor, Dr. Catherine Sherrington, I am delighted to inform you that your manuscript entitled "Evaluation of a very brief pedometer-based physical activity intervention delivered in NHS Health Checks in England: The VBI randomised controlled trial" (PMEDICINE-D-19-02857R2) has been accepted for publication in PLOS Medicine. 

PRODUCTION PROCESS

PRESS

PROFILE INFORMATION

Thank you again for submitting the manuscript to PLOS Medicine. We look forward to publishing it. 

Best wishes, 

Richard Turner, PhD

Senior Editor 

PLOS Medicine

plosmedicine.org